# Learning of probabilistic punishment as a model of anxiety produces changes in action but not punisher encoding in the dmPFC and VTA

David S Jacobs[1], Madeleine C Allen[1,2], Junchol Park[3], Bita Moghaddam[1,2]*

[1]Department of Behavioral Neuroscience, Oregon Health & Science University, Portland, United States; [2]Department of Psychiatry, Oregon Health & Science University, Portland, United States; [3]Janelia Research Campus, Howard Hughes Medical Institute, Ashburn, United States

*For correspondence:
bita@ohsu.edu

**Competing interest:** The authors declare that no competing interests exist.

**Abstract** Previously, we developed a novel model for anxiety during motivated behavior by training rats to perform a task where actions executed to obtain a reward were probabilistically punished and observed that after learning, neuronal activity in the ventral tegmental area (VTA) and dorsomedial prefrontal cortex (dmPFC) represent the relationship between action and punishment risk (Park and Moghaddam, 2017). Here, we used male and female rats to expand on the previous work by focusing on neural changes in the dmPFC and VTA that were associated with the learning of probabilistic punishment, and anxiolytic treatment with diazepam after learning. We find that adaptive neural responses of dmPFC and VTA during the learning of anxiogenic contingencies are independent from the punisher experience and occur primarily during the peri-action and reward period. Our results also identify peri-action ramping of VTA neural calcium activity, and VTA-dmPFC correlated activity, as potential markers for the anxiolytic properties of diazepam.

## Editor's evaluation

Punishment is key form of learning and behavior change, yet its core behavioral and brain mechanisms remain poorly understood and certainly less understood relative to reward learning. This manuscript uses dual fiber photometry to make an important advance in understanding how punishment is learned by studying how punishment changes action and punisher coding in the medial prefrontal cortex and ventral tegmental area of rats. The authors interpret the results as supporting a role for both areas in foraging in the face of risky outcomes. This work follows nicely on prior work and presents a straightforward and interesting experiment, using a validated anxiolytic to test what components of the neural response are related to this emotional component.

## Introduction

Anxiety is a debilitating symptom of most mental health disorders. Laboratory studies into the neural underpinnings of anxiety typically assess innate anxiety through exposure to an ambiguous context (*Lezak et al., 2017*). While this approach has provided valuable insight into the mechanisms which underlie acute fear and anxiety states, they do not address real-life situations where anxiety develops because actions are learned to be associated with a risk of an aversive outcome. An alternative approach to study this form of anxiety is to utilize uncertain punishment contingencies during action execution, which produce self-reported elevations in anxiety in humans and have been validated with

anxiolytic treatment with benzodiazepines in rodents and non-human primates (*Vogel et al., 1971*; *Fischer et al., 2010*; *Schmitz and Grillon, 2012*). Recent procedures assessing reward and punishment learning simultaneously have further indicated that learning of punishment contingencies may be critical to understanding behavioral and neuronal changes related to anxiety (*Jean-Richard-Dit-Bressel et al., 2021a*; *Jean-Richard-Dit-Bressel et al., 2019*).

In our previous work (*Park and Moghaddam, 2017*), we assessed this mode of learned anxiety by developing a punishment risk task (PRT) in rats where actions executed to obtain reward conflicted with the presence of a low probability of harm (low intensity footshock). We observed that, after PRT training, response time during risky actions became longer and more variable. Consistent with the relevance of PRT to anxiety, treatment with the anxiolytic drug diazepam diminished the impact of footshock probability on response time. To begin to understand the neural mechanisms that support PRT performance, we recorded from the dorsomedial prefrontal cortex (dmPFC) and the ventral tegmental area (VTA) over PRT training sessions. The focus on these interconnected regions was because they have been implicated in anxiety (*Eysenck et al., 2007*; *Holmes and Wellman, 2009*; *Park et al., 2016*; *Balderston et al., 2017a*; *Balderston et al., 2017b*; *Roberts, 2020*; *Jacobs and Moghaddam, 2021*) and are critical for execution of reward-guided actions (*Watabe-Uchida et al., 2017*; *Balleine and Dickinson, 1998*; *Ellwood et al., 2017*; *Flores-Dourojeanni et al., 2021*). We found that dmPFC and VTA neurons flexibly, and in coordination, represent the relationship between action and punishment. Here, we sought to address two outstanding questions: (1) Do these brain regions 'learn' about punishment contingencies by changing their responses to the punishment or reward during learning? (2) After learning, how are behavioral changes in response to anxiolytic treatment with diazepam represented in these regions?

We used fiber photometry instead of single unit recording used in the original study to (indirectly) assess the activity of neural population states in the dmPFC and VTA during PRT acquisition so that we could measure changes in the phasic neural response to the punishment (footshock) itself during task learning. After learning, we examined the effect of diazepam on dmPFC and VTA activity in correlation with behavior. We find that initial learning of probabilistic punishment changes neural calcium responses in the VTA and dmPFC during action execution and reward whereas response to the punisher remained unchanged. Moreover, anxiolytic treatment enhanced VTA peri-action activity and promoted correlated activity between the two regions without influencing dmPFC peri-action activity or encoding of the punisher.

## Methods
### Subjects
Male and female Long-Evans (bred in house n=8) and Sprague-Dawley (Charles River n=5) rats were used. Animals were pair-housed on a reverse 12 hr:12 hr light/dark cycle. All experimental procedures and behavioral testing were performed during the dark (active) cycle. All studies included both strains of male (n=7) and female (n=6) rats. All experimental procedures were approved by the OHSU Institutional Animal Use and Care Committee and were conducted in accordance with National Institutes of Health Guide for the Care and Use of Laboratory Animals.

### Surgery
#### Viral infusion surgery
Prior to task training, animals were injected with AAV8-hSyn-GCaMP6s-P2A-tdTomato (OHSU Vector Core, 5e13 ng/ml) to allow for pan-neuronal expression of fluorescent calcium indicator GCaMP6s and red fluorophore tdTomato. The coexpression of tdTomato allows for a motion artifact control signal to be used to correct GCaMP signals in rodents (*Babayan et al., 2018*; *Matias et al., 2017*; *Menegas et al., 2018*; *Soares et al., 2016*). Rats were anesthetized with isoflurane and placed in a stereotaxic apparatus. A small incision on the scalp was performed. The skull was cleaned and two craniotomies over the dmPFC and ipsilateral VTA were performed. A microinfusion syringe (Hamilton) was filled with virus and lowered into the dmPFC (AP +3.0, ML +0.6, DV –3.3 mm from dura) or VTA (AP –5.4, ML +0.6, DV –7.5 from dura) and injected into the brain at a volume of 700 nl at 50 nl/min using a syringe pump (World Instruments). Following infusion, virus was given 12 min to diffuse before the needle was

slowly removed. The incision was then closed and covered with triple antibiotic. Animals were given 5 mg/kg of carpofen after surgery and allowed at least 1 week to recover from surgery.

## Fiber implant surgery

After allowing at least 7 weeks for virus expression, subjects were implanted with an optical fiber aimed at the dmPFC, specifically the prelimbic region, (dmPFC; AP +3.0, ML +0.6, DV –3.3 mm from dura) and VTA (AP –5.4, ML +0.6, DV –7.5 mm from dura) using surgical procedures outlined in the virus infusion section, with the exception that three additional bore holes were made for skull screws and fibers were secured to the skull using a light curing dental cement (Ivoclear Vivadent). Animals were given 1 week to recover from surgery before behavioral testing.

## Initial training and PRT

The PRT follows previously published methods (*Park and Moghaddam, 2017*; *Chowdhury et al., 2019*). Rats were trained to make an instrumental response to receive a 45 mg sugar pellet (BioServe) under fixed ratio one schedule of reinforcement (FR1). The availability of the nosepoke for reinforcement was signaled by a 5 s tone. After at least three FR1 training sessions, PRT sessions began. PRT sessions consisted of three blocks of 30 trials each. The action-reward contingency remained constant, with one nosepoke resulting in one sugar pellet. However, there was a probability of receiving a footshock (300 ms electrical footshock of 0.3 mA) after the FR1 action, which increased over the blocks (0%, 6%, or 10% in blocks 1, 2, and 3, respectively). To minimize generalization of the action-punishment contingency, blocks were organized in an ascending footshock probability with 2 min timeouts between blocks. Punishment trials were pseudo-randomly assigned, with the first footshock occurring within the first five trials. All sessions were terminated if not completed in 180 min.

## Diazepam treatment

Injectable diazepam (Pfizer/Hospira, Lake Forest, IL) at a dose of 2.0 mg/kg or sterile saline (0.9% NaCl) were injected intraperitoneally 5 min before the start of the task. Injections were given in a repeated design where each subject first received saline in a session and then received diazepam in the next or the following day. All injections were given at a volume of ≤1.0 ml/kg and after at least 3 days of training.

## Fiber photometry system and recording procedures

Recordings were performed with a commercially available fiber photometry system, Neurophotometrics Model: FP3001 (NPM). Recording was accomplished by providing both 470 and 560 nm excitation light through the 400 µm core patchcord to the dmPFC or VTA for GCaMP6s and tdTomato signals, respectively. Data were recorded using bonsai open source software (*Lopes et al., 2015*) and timestamps of behavioral events were collected by 5 V TTL pulses that were relayed to an Arduino interfaced with bonsai software.

## Fiber photometry analysis

### Peri-event analysis

Signals from the 465 (GCaMP6s) and 560 (tdTomato) streams were processed in Python (version 3.7.4) using custom written scripts similar to previously published methods (*Jacobs and Moghaddam, 2020*). Briefly, 465 and 560 streams were lowpass filtered at 3 Hz using a Butterworth filter and subsequently broken up based on the start and end of a given trial. The 560 signal was fitted to the 465 using a least-squares first-order polynomial and subtracted from 465 signal to yield the change in fluorescent activity ($\Delta F/F$=465 signal − fitted 560 signal/fitted 560 signal). Peri-event z-scores were computed by comparing the $\Delta F/F$ after the behavioral action to the 4–2 s baseline $\Delta F/F$ prior to a given epoch. To investigate potential different neural calcium responses to receiving the footshock vs. anticipation, punished (i.e. shock) trials and unpunished trials were separated. Trials with a z-score value >40 were excluded. From approximately 3000 trials analyzed, this occurred on <1% of trials.

### AUC analyses

To represent individual data we calculated the area under the curves (AUCs) for each subject. To quantify peri-cue and peri-action changes we calculated a change or summation score between 1 s

before (pre event) and 1 s after (post event) cue onset or action execution. For the reward period we calculated a change score by comparing 2 s after reward delivery to the 1 s prior to reward delivery. For punished trials, response to footshock was calculated as the change from 1 s following footshock delivery compared to the 1 s before footshock. Outliers were removed using GraphPad Prism's ROUT method (Q=1%; *Motulsky and Brown, 2006*) which removed only three data points from analysis.

### Time lagged cross-correlation analysis

Cross-correlation analysis has been used to identify networks from simultaneously measured fiber photometry signals (*Sych et al., 2019*). For rats with properly placed fibers in the dmPFC and VTA, correlations between photometry signals arising in the VTA and dmPFC were calculated for the peri-action, peri-footshock, and peri-reward periods using the z-score normalized data. The following equation was used to normalize covariance scores for each time lag to achieve a correlation coefficient between –1 and 1:

$$\mathrm{Coef} = \mathrm{Cov}/(s^{1*}s^{2*}n)$$

where Cov is the covariance from the dot product of the signal for each time point, $s^1$ and $s^2$ are the standard deviation of the dmPFC and VTA streams, respectively, and n is the number of samples. An entire cross-correlations function was derived for each trial and epoch.

Comparison to electrophysiology results: Fiber photometry data for the third PRT session were compared to the average of the 50 ms binned single unit data (see Figure 4 of *Park and Moghaddam, 2017*). This third PRT session corresponds to the session electrophysiology data were collected from. To overlay data from the two techniques, data were lowpass filtered at 3 Hz and photometry data were downsampled to 20 Hz (to match the 50 ms binning). Data from both streams were then min-max normalized between 0 and 1 at the corresponding cue and action+reward epochs.

To assess the similarity of the two signals, we performed a Pearson correlation analysis between the normalized single unit and fiber photometry data for cue or action+reward epochs at each risk block, as well as between randomly shuffled photometry signals with single unit response as a control. For significant Pearson correlations we performed cross-correlation analysis (see above) to investigate if the photometry signal lagged behind electrophysiology given the slower kinetics of GCAMP6 compared to single unit approaches (*Chen et al., 2013*).

## Statistical analysis

For FR1 training, trial completion was measured as the number of food pellets earned. Data were assessed for the first three to four training sessions. Action and reward latencies were defined as time from cue onset to action execution or from food delivery until retrieval, respectively. Values were assessed using a mixed effects model with the training as a factor and post hoc tests were performed using the Bonferroni correction where appropriate.

For the PRT, trial completion was measured as the percentage of completed trials (of the 30 possible) for each block. Action latencies were defined as time from cue onset to action execution. Data were analyzed using a two-way RM ANOVA or mixed effects model. Because there were missing data for non-random reasons (e.g. failure to complete trials in response to punishment risk) we took the average of risk blocks (blocks 2 and 3) and the no risk block (block 1) to permit repeated measures analysis. We used mixed effects model if data were missing for random reasons. Risk and session were used as factors and post hoc tests were performed using the Bonferroni correction where appropriate. When only two groups were compared a paired t-test or Wilcoxon test was performed after checking normality assumption through the Shapiro-Wilk test.

To assess changes in neural calcium activity, we utilized a permutation-based approach as outlined in *Jean-Richard-Dit-Bressel et al., 2020* using Python (version 3). An average response for each subject for a given time point in the cue, action, or reward delivery period was compared to either the first PRT or saline session. For each time point a null distribution was generated by shuffling the data, randomly selecting the data into two groups, and calculating the mean difference between groups. This was done 1000 times for each time point and a p-value was obtained by determining the percentage of times a value in the null distribution of mean differences was greater than or equal to the observed difference in the unshuffled data (one-tailed for comparisons to 0% risk and FR1 data, two-tailed for all other comparisons). To control for multiple comparisons we utilized a consecutive

threshold approach based on the 3 Hz lowpass filter window (*Jean-Richard-Dit-Bressel et al., 2020*; *Pascoli et al., 2018*), where a p-value <0.05 was required for 14 consecutive samples to be considered significant.

To assess AUC changes in photometry data, we compared all risk blocks and all sessions using ANOVA with factors risk block and session. Because not all subjects completed learning and diazepam data, we used an ordinary two-way ANOVA. Significant main effects and interactions were assessed with post hoc Bonferroni multiple comparison tests.

To assess correlated activity changes as a function of risk or session, we took the peak and 95% confidence interval for the overall cross-correlation function. These values were compared by a two-way ANOVA with factors risk and session and utilized a post hoc Bonferroni correction.

Other than permutation tests, all statistical tests were done using GraphPad Prism (version 8) and an α of 0.05. Results for all statistical tests and corresponding figures can be found in *Table 1* or supplemental figures.

### Excluded data

Outliers from latency analysis were removed when a data point was >5 SDs above the mean across all blocks. This removed one data point from analysis. In FR1 studies, data from one rat's third and fourth session were excluded because the camera became misaligned with the patch cord and thus the last (fifth) FR1 session was used for analysis. In PRT studies, data from the dmPFC of one session for a rat was excluded due to lack of timestamp collection and one block of a session was excluded for two other rats because the control 560 nm LED failed for the dmPFC.

Four rats with VTA placement were excluded because fibers were placed outside the VTA or GCaMP6s expression was not observed. Several rats did not complete all phases of the experiment due to lost fiber implants, leaving the final sample sizes as n=9 and n=7 for dmPFC in learning and diazepam treatment stages, respectively, and n=4 for VTA in learning and diazepam treatment stages.

### Histology and imaging

Viral expression and fiber placements were verified after behavioral testing. Subjects were transcardially perfused with 0.01 M phosphate buffered saline (PBS) followed by 4% paraformaldehyde (PFA). Brains were removed and post-fixated in PFA for 24 hr before being placed in 20% sucrose solution and stored at 4°C. Forty μm brain slices were collected on a cryostat (Leica Microsystems) and preserved in 0.05% phosphate buffered azide. Brain slices were mounted to slides and cover slipped with Vectashield anti-fade mounting medium (Vector Labs). A Zeiss Apotome.2 microscope was used to image brain slices for GFP (Zeiss Filter set 38: 470 nm excitation/525 nm emission) and tdTomato (Zeiss Filter Set 43: 545 nm excitation/605 nm emission) to validate expression of both fluorophores in cells near the fiber tip. Fiber placement was determined by the brain slice demonstrating the most ventral fiber location.

Immunohistochemistry with a GFP antibody was used if a subject lacked virus expression to confirm the presence or absence of GCaMP6s. Brain slices were permeabilized in 3% BSA, 0.5% Triton-X, and 5% Tween 80 dissolved in PBS + 0.05% sodium azide for 2 hr at room temperature. Slices were then incubated with rabbit antiserum against GFP (Abcam, Catalogue# 6556, 1:500) diluted in PBS + Azide, 3% BSA + 0.1% Triton, and 1% Tween for 48 hr at 4°C. Slices were then washed in PBS + Azide, 3% BSA + 0.1% Triton + 1% Tween, three times for 5 min each. After this, slices were incubated with goat-anti-rabbit Alexa-488 (Abcam, Catalogue# 150081, 1:2000) diluted in PBS + Azide, 3% BSA + 0.1% Triton, and 1% Tween for 24 hr at 4°C and subsequently washed again as outlined above. Slices were then mounted to slides with Vectashield and imaged using the same procedures outlined above.

## Results

### Learning of PRT

To determine if learning of the PRT is associated with changes in behavior and neural activity, we recorded neural calcium activity in the dmPFC and VTA using fiber photometry during the first three sessions of PRT training. Fibers were generally placed in the prelimbic region of the mPFC and slightly lateral portion of the VTA (*Figure 1—figure supplement 1*). Task training was as described before (*Park and Moghaddam, 2017*; *Chowdhury et al., 2019*). After rats learned to execute an action to

**Table 1.** Statistical results for behavior and correlation/cross-correlation analyses.

| Relevant Figure(s) | Test | Factor(s): F(dfn,dfd) | p-value | Post-hoc tests (Bonferroni corrected) |
|---|---|---|---|---|
| 1C-right and S1-2A | Two-way RM ANOVA (greenhouse-geisser corrected) | Risk: F(1,7)=13.6 Session F(1.5,10.2)=5.12 Risk*Session: F(1,5,10.2)=5.12 | 0.008 0.037 0.038 | No risk trial completion not compared; all subjects completed 100% Wilcoxon(1 vs 2): W=-28,P=.032,two-tailed paired t(2 vs 3): t(8)=0.41,P>.69, two-tailed |
| 1D-right and S1-2B | Two-way RM ANOVA (greenhouse-geisser corrected) | Risk: F(1,7)=20.3 Session: F(1.5,10.2)=3.5 Risk*Session: F(1,5,10.2)=2.8 | 0.003 0.07 0.11 | None, significant effect had only 2 levels- Risk blocks >No Risk |
| 1E-right and S1-2C | Two-way RM ANOVA (greenhouse-geisser corrected) | Risk: F(1,7)=1.35 Session: F(1.97,13.8)=2.85 Risk*Session: F(1.1,7.7)=4.31 | 0.28 0.09 0.07 | None |
| 3A | One-way mixed effects | Training day: F(3,23)=15.4 | <0.001 | Bonferroni test (1 vs 2): t=3.7, P=.0038, two-tailed Wilcoxon (1 vs 3): W=36, P=.024, two-tailed Wilcoxon (1 vs 4): W=36, P=.024, two-tailed |
| 3B | One-way mixed effects | Training day: F(3,23)=10.0 | 0.0002 | Bonferroni test (1 vs 2): t=3.8, P=.003, two-tailed Bonferroni test (1 vs 3): t=4.7, P=.0003, two-tailed Bonferroni test (1 vs 4): t=4.6, P=.0004, two-tailed |
| 3C | One-way mixed effects | Training day: F(3,23)=6.58 | 0.002 | Wilcoxon (1 vs 2): W=-31, P=.21, two-tailed Wilcoxon (1 vs 3): W=-45, P=.012, two-tailed Wilcoxon (1 vs 4): W=-36, P=.024, two-tailed |
| 3E | One-way mixed effects | Training day: F(3,22)=3.2 | 0.044 | Bonferroni test (1 vs 2): t=2.2, P=.11, two-tailed Bonferroni test (1 vs 3): t=2.2, P=.12, two-tailed Bonferroni test (1 vs 4): t=2.9, P=.03, two-tailed |
| 3F | One-way mixed effects | Training day: F(3,22)=3.2 | 0.045 | Bonferroni test (1 vs 2): t=0.3, P=.99, two-tailed Bonferroni test (1 vs 3): t=2.1, P=.16, two-tailed Bonferroni test (1 vs 4): t=2.55, P=.05, two-tailed |
| 3H | One-way mixed effects | Training day: F(3,7)=.12 | 0.94 | |
| 3I | One-way mixed effects | Training day: F(3,7)=4.74 | 0.041 | Bonferroni test (1 vs 2): t=1.24, P=.76, two-tailed Bonferroni test (1 vs 3): t=2.89, P=.07, two-tailed Bonferroni test (1 vs 4): t=3.25, P=.042, two-tailed |
| 3-S1A | RM one-way ANOVA (greenhouse-geisser corrected) | Trial number: F(1.9,15.1)=2.24 | 0.14 | |
| 3-S1D | RM one-way ANOVA (greenhouse-geisser corrected) | Trial number: F(1.4,3.4)=1.4 | 0.31 | |

*Table 1 continued on next page*

*Table 1 continued*

| Relevant Figure(s) | Test | Factor(s): F(dfn,dfd) | p-value | Post-hoc tests (Bonferroni corrected) |
|---|---|---|---|---|
| 5A-right, S5-1A | Two-way RM ANOVA (greenhouse-geisser corrected) | Risk: F(1,6)=1<br>Session: F(1,6)=1<br>Risk*Session: F(1,6)=1 | 0.36<br>0.36<br>0.36 | None |
| 5B-right, S5-1B | Two-way RM ANOVA (greenhouse-geisser corrected) | Risk: F(1,6)=1.4<br>Session: F(1,6)=1.3<br>Risk*Session: F(1,6)=6.65 | 0.28<br>0.29<br>0.041 | Wilcoxon (no risk saline vs. no risk diazepam): W=22, p=0.08 (one-tailed)<br>Wilcoxon (risk saline vs. risk diazepam): W=−24, p=0.047 (one-tailed) |
| 5C-right, S5-1C | Two-way RM ANOVA (greenhouse-geisser corrected) | Risk: F(1,6)=19.15<br>Session: F(1,6)=6.5<br>Risk*Session: F(1,6)=4.8 | 0.004<br>0.04<br>0.07 | For risk: Only two levels – no risk>risk<br>For session: Only two levels – diazepam>saline |
| 7-S2/3A | Two-way ordinary ANOVA (dmPFC cue) | Risk: F(2,94)=2.4<br>Session: F(4,94)=0.65<br>Risk*Session: F(8,94)=0.98 | 0.10<br>0.65<br>0.46 | |
| 7-S2/3B | Two-way ordinary ANOVA (dmPFC action) | Risk F(2,94)=7.4<br>Session F(4,94)=0.91<br>Risk*Session F(8,94)=0.51 | 0.001<br>0.46<br>0.85 | Bonferroni test (0% vs. 6%): t=2.2, p=0.03, one-tailed<br>Bonferroni test (0% vs. 10%): t=3.8, p=0.00025, one-tailed |
| 7-S2/3C | Two-way ordinary ANOVA (dmPFC reward) | Risk: F(2,93)=9.18<br>Session: F(4,93)=1.77<br>Risk*Session: F(8,93)=0.54 | <0.001<br>0.14<br>0.83 | Bonferroni test (0% vs. 6%): t=0.32, p=0.99, two-tailed<br>Bonferroni test (0% vs. 10%): t=3.9, p≤0.001, two-tailed |
| 7-S2/3D | Two-way ordinary ANOVA (dmPFC footshock) | Shock: F(1,68)=78.7<br>Session: F(4,68)=1.46<br>Shock*Session: F(4,68)=0.81 | <0.001<br>0.23<br>0.52 | None required: Shock >No shock |
| 7-S2/3E | Two-way ordinary ANOVA (VTA cue) | Risk: F(2,42)=14.8<br>Session: F(4,42)=1.52<br>Risk*Session: F(8,42)=2.06 | <0.001<br>0.21<br>0.06 | Bonferroni test (0% vs. 6%): t=.32, p=0.99, two-tailed<br>Bonferroni test (0% vs. 10%): t=4.9, p≤0.001, two-tailed |
| 7-S2F | Two-way ordinary ANOVA (VTA action change) | Risk: F(2,42)=10.5<br>Session: F(4,42)=6.1<br>Risk*Session: F(8,42)=2.43 | <0.001<br><0.001<br>0.03 | *Risk*:<br>Bonferroni test (0% vs. 6%): t=0.32, p=0.99, two-tailed<br>Bonferroni test (0% vs. 10%): t=4.2, p=0.0003, two-tailed<br>*Session* (via Bonferroni tests): All sessions greater than Session 1 (p-values <0.007), except for diazepam (p=0.11)<br>*Interaction* (via Bonferroni tests): No differences between sessions at 0% or 6% risk (p-values >0.10). At 10% risk Sessions 2 and 3 greater than Session 1 (p-values <0.023). Saline and diazepam not different from Session 1 (p-values >0.13). |

*Table 1 continued*

| Relevant Figure(s) | Test | Factor(s): F(dfn,dfd) | p-value | Post-hoc tests (Bonferroni corrected) |
|---|---|---|---|---|
| 7-S3F | Two-way ordinary ANOVA (VTA action sum) | Risk: F(2,40)=1.67 Session: F(4,40)=5.87 Risk*Session: F(8,40)=2.40 | 0.20 <0.001 0.03 | *Session* (via Bonferroni tests): Diazepam greater than Session 1 (p=0.0003), all other session not significantly different (p-values >0.07) *Interaction* (via Bonferroni tests): 0% risk Diazepam greater than Session 1 (p=0.004), other sessions not significant (p-values = 0.99). 6% risk Diazepam greater than Session 1 (p=0.002), other sessions not significant (p-values >0.22). 10% risk all sessions same as session 1 (p-values >0.18). |
| 7-S2/3G | Two-way ordinary ANOVA (VTA reward) | Risk: F(2,42)=3.47 Session: F(4,42)=3.12 Risk*Session: F(8,42)=0.69 | 0.04 0.025 0.69 | *Risk*: Bonferroni test (0% vs. 6%): t=1.4, p=0.34, two-tailed Bonferroni test (0% vs. 10%): t=2.63, p=0.024, two-tailed *Session* (via Bonferroni tests – two tail): Session 3 greater than session 1 (p=0.049), all other sessions not significant (p-values >0.68). |
| 7-S2/3H | Two-way ordinary ANOVA (VTA footshock) | Shock: F(1,30)=36.9 Session: F(4,30)=1.55 Shock*Session: F(4,30)=0.28 | <0.001 0.21 0.89 | None required: Shock > No shock |
| 8D | Two-way ordinary ANOVA | Session: F(1,596)=27.66 Risk block: F(2,596)=21.85 Risk*Session: F(2,596)=7.902 | <0.0001 <0.0001 0.0004 | Saline vs. diazepam (Bonferroni test – two tail): 0% risk: p≤0.001 6% risk: p≤0.001 10% risk: p=0.99 |
| 8G | Two-way ordinary ANOVA | Session: F(1,596)=82.63 Risk block: F(2,596)=14.55 Risk*Session: F(2,596)=9.124 | <0.0001 <0.0001 0.0001 | Saline vs. diazepam (Bonferroni test – two tail): 0% risk: p≤0.001 6% risk: p≤0.001 10% risk: p=0.238 |

| Relevant Figure(s) | Test | Pearson r | p-value (cue) | p-value (action/reward) |
|---|---|---|---|---|
| S2-4C | Pearson correlation | Cue-absolute r values <0.26 Action/reward- absolute r values >0.56 | p-values >0.09 | Action/reward – p-values <0.001 |
| S2-4D | Pearson correlation | Cue-absolute r values <0.08 Action/reward- absolute r values >0.72 | p-values >0.63 | Action/reward – p-values <0.001 |

receive a reward, a (varying) risk of shock was introduced contingent on the action (*Figure 1A*). The risk of footshock increased from 0% to 10% during three 30-trial consecutive blocks (*Figure 1B*). During the first three sessions of PRT, increases in the latency to execute to the risky action and decreases in trial completion were observed (*Figure 1C–D* – left, *Table 1*). In Session 1, punishment risk appeared non-specific because some subjects showed increases in the latency to retrieve reward (*Figure 1E*). Over PRT training suppression of action execution was observed specifically in risk blocks and did not change between Sessions 2 and 3 (*Figure 1C–D* – right, *Table 1*). Furthermore, reward retrieval increases from risk of footshock were attenuated in later sessions, although this effect was not statistically significant (*Figure 1E*- – right, *Table 1*). No behavioral changes were seen during PRT learning in the safe block (*Figure 1—figure supplement 2*).

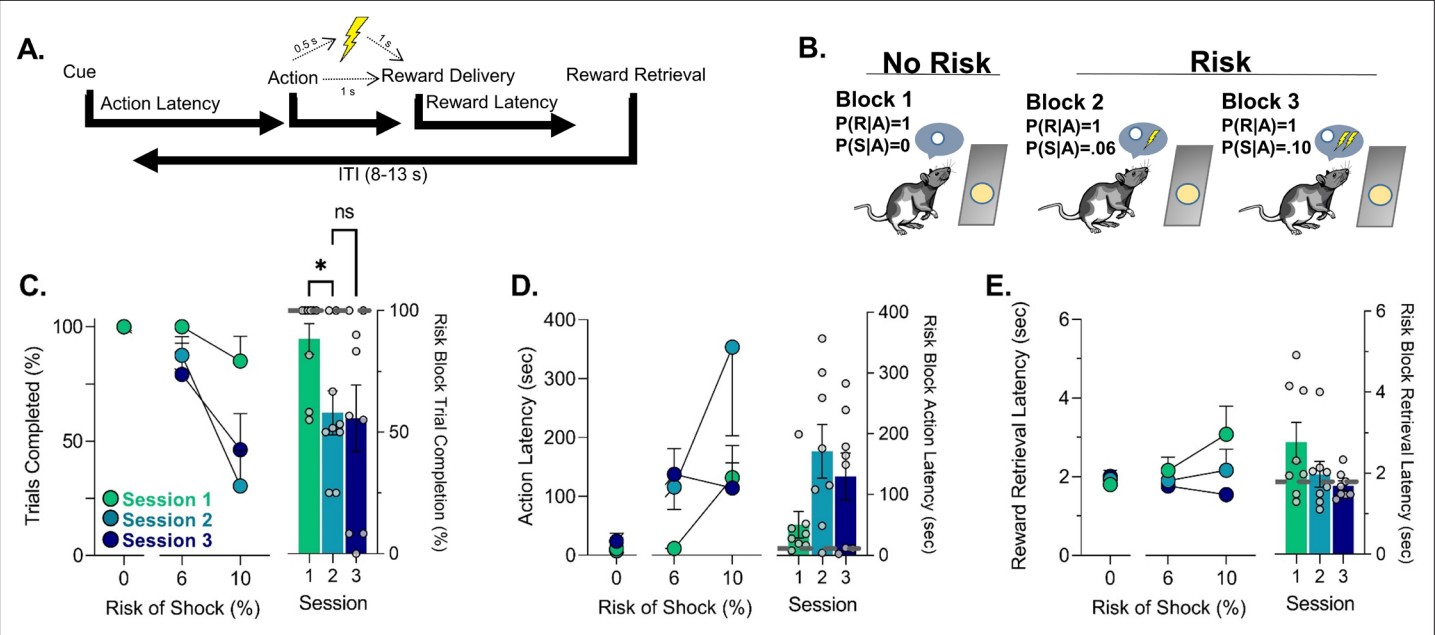

**Figure 1.** Schematic of punishment risk task (PRT) design and PRT behavior. (**A**) Outline of trial structure in the PRT where action led to reward delivery with a varying risk of footshock. (**B**) The multi-component schedule used to probabilistically punish the action with ascending risk of footshock. (**C**) Trial completion over the first three sessions in each block and when comparing risk blocks over those sessions (right-side bar plot). (**D**) Changes in latency to action completion over the first three sessions and specifically in risk blocks (right-side bar plot). (**E**) Latency to retrieve the food reward over the first three sessions and specifically in risk blocks (right-side bar plot). Gray lines on bar plots indicate the average for the safe block in Session 1 (i.e. before punishment was ever encountered). Data are presented as mean ± SEM with small dots reflecting individual subjects.*p<0.05, ns = not significant, for exact p-values see **Table 1**. n=8–9 rats.

The online version of this article includes the following source data and figure supplement(s) for figure 1:

**Source data 1.** Source data for **Figure 1** and **Figure 1—figure supplements 1–2**.

**Figure supplement 1.** Hit map and representative image of GFP-GCaMP6s and tdTomato expression in the dorsomedial prefrontal cortex (dmPFC) and ventral tegmental area (VTA).

**Figure supplement 2.** Average (mean ± SEM) and individual (gray dots) behavior during the safe (0% risk) block for the first three learning sessions.

## dmPFC and VTA dynamically represent actions in the PRT

Next, we compared dmPFC and VTA neural calcium activity in Sessions 1–3 to determine if neural responses to task events change in these regions during PRT training. Data shown in **Figure 2** and Figure 4 compare these responses in each block. For each session, we separated trials which resulted in the footshock punishment from those that did not to determine if activity patterns were observed based on whether a trial was ultimately punished, or risky but ultimately safe. Specifically, we were interested in whether a change in neural response to the actual punisher would occur during learning, or if changes in neural calcium responses to task events occurred when risk was present but footshock was not received.

**Figure 2** shows data from all three blocks excluding trials in blocks 2 and 3 where a footshock was received. In block 1 (the safe block) response to all task events remained the same during PRT learning. Session-by-session differences began to emerge in risk blocks. During 6% risk blocks, the initial modest peri-action inhibitory response decreased in both regions suggesting that action-related activity is sensitive to punishment risk (**Figure 2B and E**; **Figure 2—figure supplement 1**). These peri-action responses continued to be neutral or elevated during 10% risk blocks in the dmPFC and VTA, respectively. Comparison of these data to the first (safe) block for each session indicated that there was a significant attenuation of the dmPFC peri-action response in risk blocks in Session 3 (**Figure 2—figure supplement 2C,E**). While peri-action responses began to decrease in the 10% risk block for Session 1, these changes did not reach significance (**Figure 2—figure supplement 2A,E**). Similarly, significant peri-action increases in neural calcium activity were observed in the 10% risk block for the VTA in Session 3 (**Figure 2—figure supplement 3C,E**).The learning sessions in the PRT were

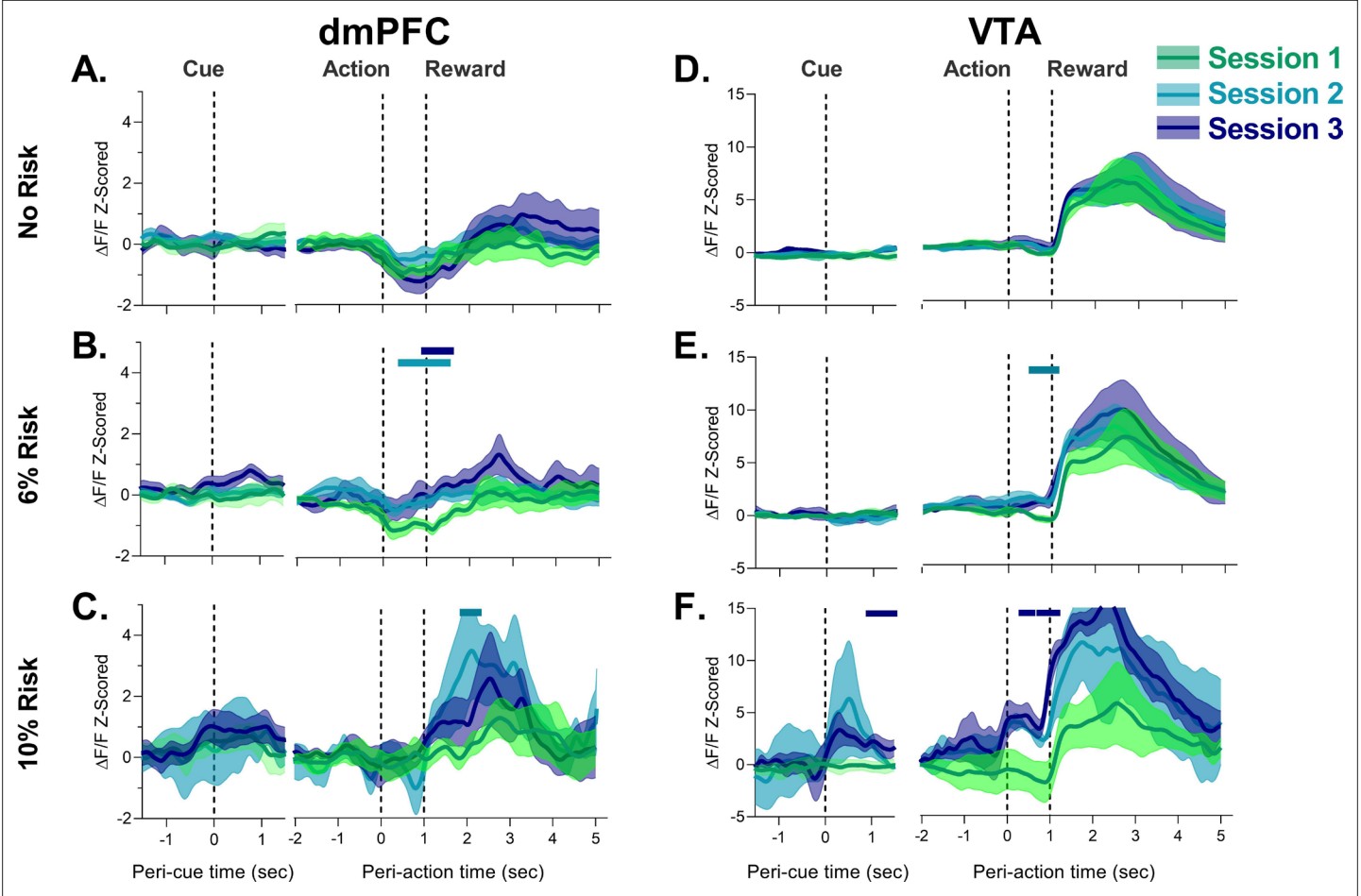

**Figure 2.** Neural calcium responses to task events in unpunished trials by dorsomedial prefrontal cortex (dmPFC) and ventral tegmental area (VTA) during punishment risk task (PRT) learning. (**A–C**) dmPFC responses to cue, action, and reward delivery for each block during the PRT task. Changes in dmPFC peri-action response were observed when risk was present. (**D–F**) VTA responses to cue, action, and reward delivery in the VTA. Changes in VTA action and cue and action responses were detected with task training. Solid bars indicate significant differences from Session 1, where the color of the bar denotes the different session. Traces represent the mean with shaded region indicating ± SEM. n=3–9 rats, n=2 rats for VTA Session 2 at 10% risk.

The online version of this article includes the following source data and figure supplement(s) for figure 2:

**Source data 1.** Source data for *Figure 2* and *Figure 2—figure supplements 2–4*.

**Figure supplement 1.** Permutation test results for recordings in the dorsomedial prefrontal cortex (dmPFC) and ventral tegmental area (VTA) for unpunished trials during punishment risk task (PRT) learning.

**Figure supplement 2.** Dorsomedial prefrontal cortex (dmPFC) neural calcium responses to task events in unpunished trials.

**Figure supplement 3.** Ventral tegmental area (VTA) neural responses to task events in unpunished trials.

**Figure supplement 4.** Comparison between mean dorsomedial prefrontal cortex (dmPFC) and ventral tegmental area (VTA) fiber photometry (green trace – current study) and electrophysiological single unit recordings (black trace – *Park and Moghaddam, 2017*) during the corresponding session and epochs in the punishment risk task (PRT).

also associated with a transient increase in dmPFC and VTA peri-reward activation (*Figure 2C and F*, *Figure 2—figure supplement 1*). These results suggest that both the VTA and dmPFC change their representation of risky actions and reward after exposure to anxiogenic contingencies.

In the VTA, a robust response to cue was observed in Session 3 (*Figure 2F*; *Figure 2—figure supplement 3*). In the dmPFC, a similar but weaker response was observed (*Figure 2—figure supplement 2C,E*). It should be noted, however, that the dmPFC and VTA responses to the cue were not observed during punished trials where there were fewer trials and thus the current approach may not be optimal for detection of cue responses (also see *Figure 2—figure supplement 4*).

To better understand the peri-event changes in neural calcium activity during PRT exposure we analyzed neural activity during FR1 learning, before any footshock punishment was experienced. Peri-action decreases in dmPFC neural calcium activity were not observed in the first FR1 session but emerged later in training when behavior became asymptotic in FR1 Session 4 (*Figure 3A–D*). AUC analysis indicated that dmPFC changes during action execution and reward were attenuated as animals learned the FR1 schedule (*Figure 3E–F*, see *Table 1* for statistics). To assess whether prior PRT data could be related to time engaged in session (as opposed to punishment risk), we assessed whether the phasic decrease in FR1 Session 4 changed within the session. This was not the case as the phasic decrease in peri-action activity was observed in early (1–30), middle (31–60), and late (61–90) FR1 trials (*Figure 3—figure supplement 1B*). In the VTA, FR1 training was accompanied by a more rapid rise in response to reward delivery in Sessions 4, with no change in action-related activity (*Figure 3G–I*, see *Table 1* for statistics). VTA response to reward did not vary within session for Session 4 (*Figure 3—figure supplement 1C-D*).

*Figure 4* shows the session-by-session phasic response to footshock after action execution in blocks 2 and 3. Footshock produced a large increase in neuronal activity in dmPFC and VTA which reached its peak about 1 s after action and 1 s before reward delivery (*Figure 4A–B*). It is unlikely this increase is due to reward anticipation as in unpunished trials the increase in activity did not peak until about 1.5–2 s after action and comparing punished trials to unpunished trials revealed footshock-related increases were seen significantly earlier than food delivery increases (*Figure 4—figure supplement 1*). The magnitude of this increase in the footshock/pre-reward period did not change in either region as animals learned the PRT (*Figure 4—figure supplement 2*), suggesting that learning of action-punishment contingency is not associated with increases or decreases in neuronal response to the punisher itself. Responses to task events at no risk or after footshock were not observed in animals where fiber placement was outside of the VTA with no GCaMP6s expression (*Figure 4—figure supplement 3*).

Inhibitory peri-action activity in the dmPFC and increased peri-reward activity in the VTA are consistent with a large body of unit recording data (*Mulder et al., 2003*; *Simon et al., 2015*; *Park and Moghaddam, 2017*). To more directly assess the relationship between the present photometry responses and single unit activity at the corresponding session of PRT, we compared the current data with our single unit recording results (in *Park and Moghaddam, 2017*). Both dmPFC and VTA photometry signals were positively correlated with population averaged single unit recordings at a higher level than when photometry data was randomly shuffled, mostly in the action and reward period (*Figure 2—figure supplement 4*). Relatedly, significant correlations were only observed in the action and reward period and not the cue period (*Figure 2—figure supplement 4C-D*; *Table 1*). The lack of detection of the faster (<1 s) cue responses by fiber photometry, particularly in some of the VTA data, compared to the more prolonged (>1 s) signal changes in action and reward periods may be related to GCaMP6s' slower kinetics compared to unit approaches. Inspection of traces suggests that phasic responses were occasionally slower to peak in photometry data (*Figure 2—figure supplement 4*, see E) and slower to begin to decay; the latter of which could also account for the low correlation seen in some of the faster epochs such as the cue. Overall, these results suggest that fiber photometry captures some aspects of overall population activity seen in these regions during action and reward in the PRT but may be more limited when comparing to faster responses like those related to cue onset.

## Effect of diazepam on behavior and dmPFC and VTA neural calcium activity during task events

After animals were trained in the PRT, we sought to determine if anxiolytic treatment changed the neural response to task events by the dmPFC and VTA during corresponding changes in behavior. Consistent with our previous results (*Park and Moghaddam, 2017*), a low dose of diazepam (2 mg/kg) attenuated the latency to execute an action when risk was present (*Figure 5B*) without influencing the number of trials completed (*Figure 5A*; *Table 1*). Qualitatively, the effect of diazepam was most pronounced at the 6% risk block (block 2) potentially because the effectiveness of this low dose began to dissipate toward the end of block 3. Risk block action latencies following diazepam treatment were significantly shorter than saline treatment (*Figure 5B* – right). Disruptive effects of diazepam were noted in the safe block where a non-significant increase in action latency was observed in some subjects and a significant increase in latency to reward retrieval was observed (*Figure 5—figure*

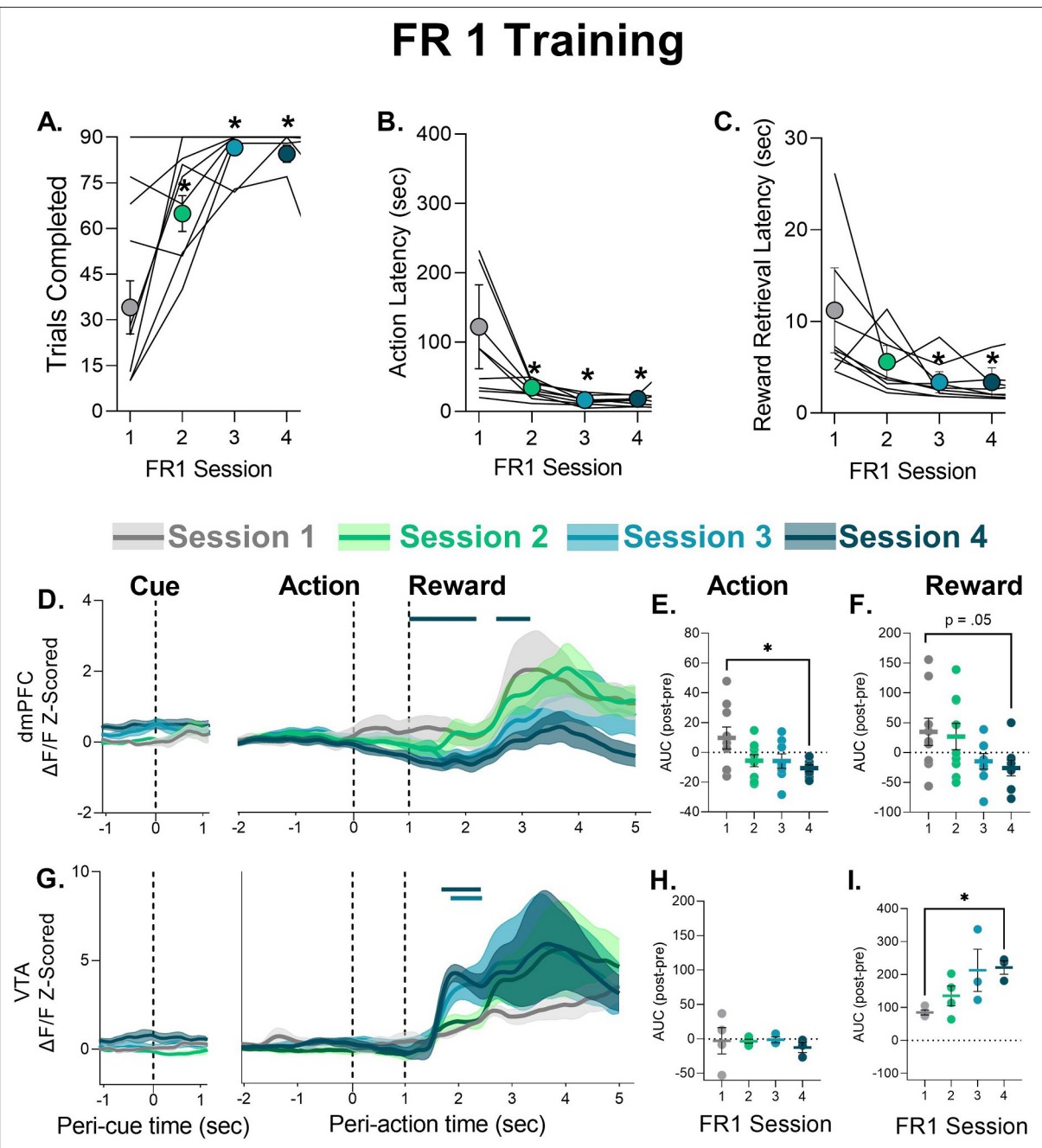

**Figure 3.** Behavior during FR1 training and corresponding neural calcium responses in cue, action, and reward epochs. (**A**) Number of trials completed. (**B**) Latency to perform the nosepoke after the tone cue. (**C**) Latency to retrieve the food pellet. Data are presented as mean ± SEM. Lines indicate individual subjects. (**D**) Mean ± SEM trace of neural calcium activity in the dorsomedial prefrontal cortex (dmPFC) for each epoch over the first four FR1 sessions. Solid bar above traces indicates significant difference from Session 1. (**E–F**) Mean ± SEM area under the curve (AUC) change scores for action and reward, respectively, with circles denoting individual subjects. (**G**) Mean ± SEM trace of neural calcium activity in the ventral tegmental area (VTA) for each epoch over the first four FR1 sessions. Solid bar above traces indicates significant difference from Session 1. (**H–I**) Mean ± SEM AUC change scores for action and reward, respectively, with circles denoting individual subjects. *p<0.05 vs. Session 1, for exact p-values see *Table 1*. n=8–9 dmPFC, n=3–4 VTA.

The online version of this article includes the following source data and figure supplement(s) for figure 3:

**Source data 1.** Source data for *Figure 3* and *Figure 3—figure supplement 1*.

**Figure supplement 1.** Response of dorsomedial prefrontal cortex (dmPFC) and ventral tegmental area (VTA) to action and reward after FR1 was learned.

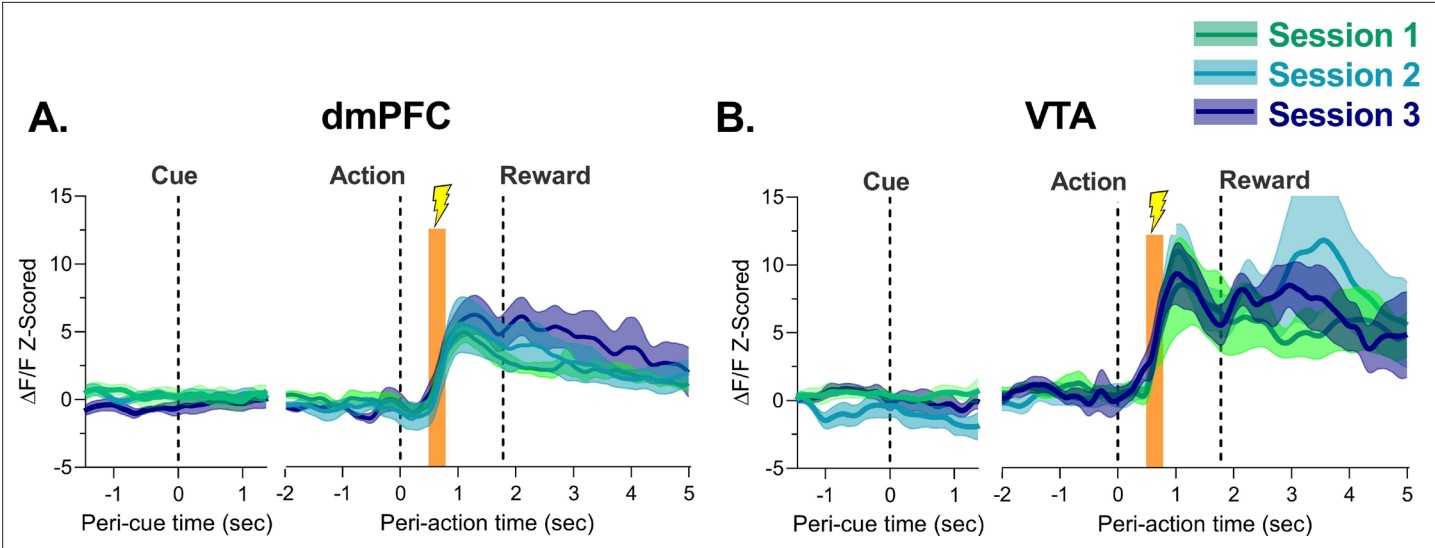

**Figure 4.** Neural calcium responses to action-contingent footshock by dorsomedial prefrontal cortex (dmPFC) and ventral tegmental area (VTA) during punishment risk task (PRT) learning. (**A**) The dmPFC demonstrated robust phasic increases in neural calcium activity at the time of footshock administration over the three initial PRT sessions. (**B**) Same as (**A**) but for the VTA. Orange bar indicates the period where footshock was administered. Traces represent mean with shaded region indicating ± SEM. n=4–9 rats.

The online version of this article includes the following source data and figure supplement(s) for figure 4:

**Source data 1.** Source data for *Figure 4* and *Figure 4—figure supplement 3*.

**Figure supplement 1.** Permutation test results for recordings in the dorsomedial prefrontal cortex (dmPFC) and ventral tegmental area (VTA) for punished trials during punishment risk task (PRT) learning.

**Figure supplement 2.** Permutation test results for recordings in the dorsomedial prefrontal cortex (dmPFC) and ventral tegmental area (VTA) for punished trials during punishment risk task (PRT) learning.

**Figure supplement 3.** Ventral tegmental area (VTA) fiber photometry traces for unpunished (0% risk block) and punished (footshock) trials during punishment risk task (PRT) learning in rats with misplaced fibers or no GCaMP6s expression.

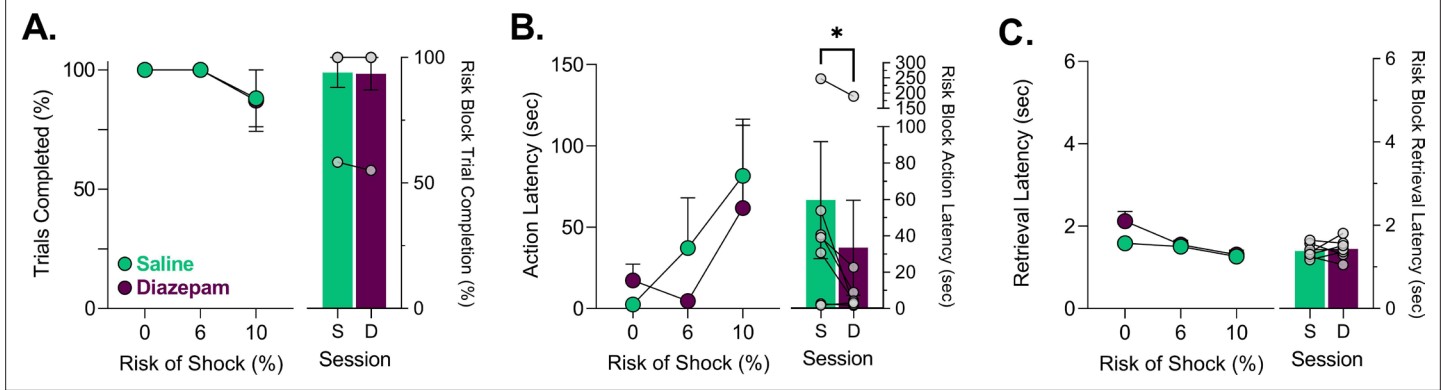

**Figure 5.** Effects of saline and diazepam (2 mg/kg) on punishment risk task (PRT) behavior. (**A**) Trial completion was unaffected by 2 mg/kg diazepam. (**B**) Action latencies for trials where risk was present or absent. Diazepam significantly and consistently attenuated action latency increases seen from probabilistic punishment only when risk was present. (**C**) Increases in reward retrieval latency which indicate motoric disruption from diazepam in block 1 dissipated in blocks 2 and 3 where risk was present. Data are presented as mean ± SEM with gray dots reflecting individual subjects. *p<0.05, for exact p-values see *Table 1*. n=7 rats. S = Saline, D= Diazepam.

The online version of this article includes the following source data and figure supplement(s) for figure 5:

**Source data 1.** Source data for *Figure 5* and *Figure 5—figure supplement 1*.

**Figure supplement 1.** Average (mean ± SEM) and individual (gray dots) behavior during the safe (0% risk) block for saline and diazepam pretreatment sessions.

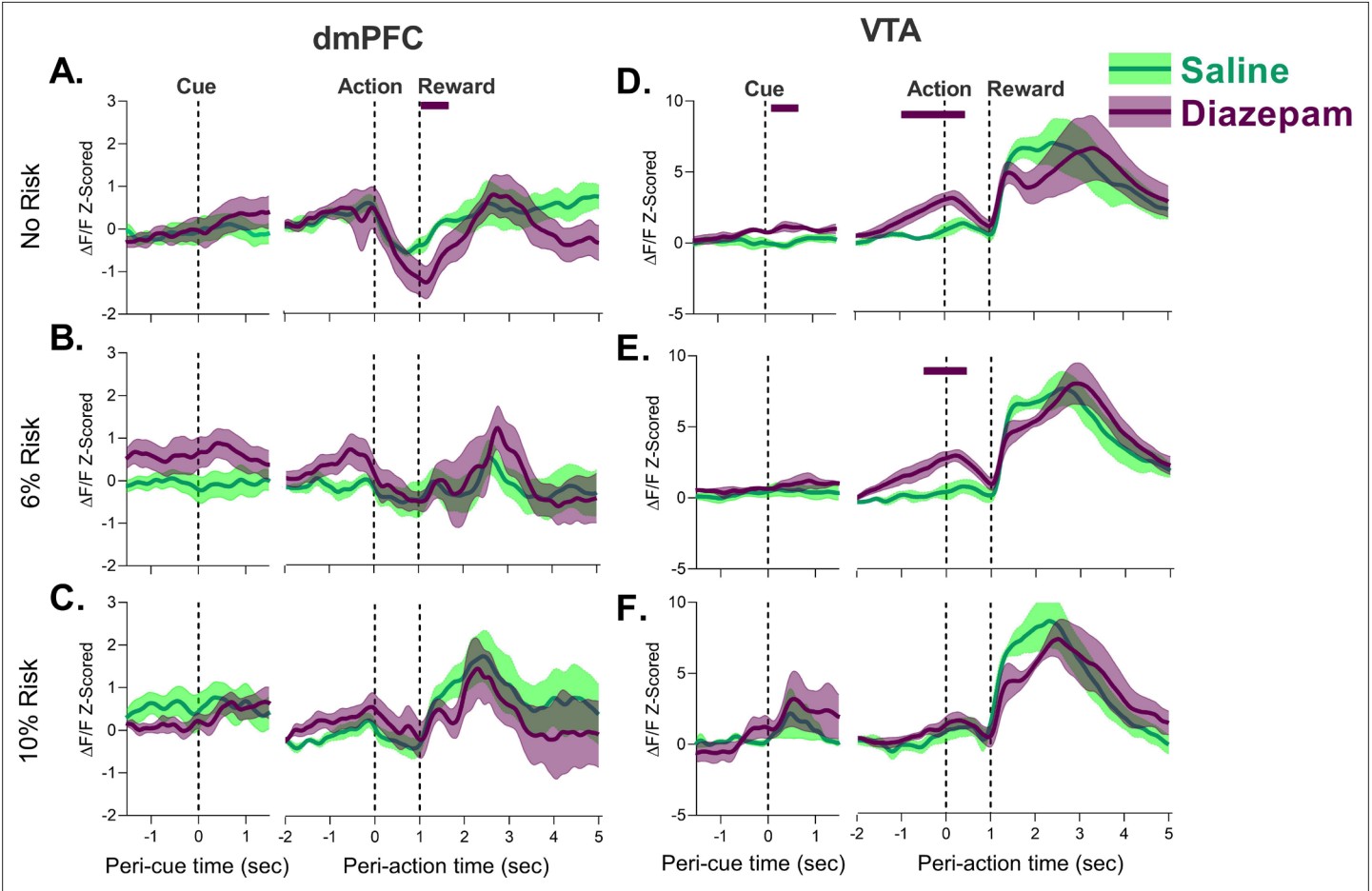

**Figure 6.** Effects of diazepam on neural calcium activity during unpunished trials in the punishment risk task (PRT) in the dorsomedial prefrontal cortex (dmPFC) (left) and ventral tegmental area (VTA) (right). (**A–C**) No effect of diazepam was observed during the cue or action epoch in the dmPFC, and a small but significant downward shift was seen following treatment early in the reward epoch in the safe block. (**D–F**) Diazepam had no effect on neural calcium activity during the reward period in the VTA. The peri-action activity was enhanced by diazepam until after action execution (**D–E**) but dissipated at high risk. (**F**) Traces represent the mean with shaded region indicating ± SEM. Solid lines above traces indicate significant differences from saline at those time points. n=4–7 rats.

The online version of this article includes the following source data and figure supplement(s) for figure 6:

**Source data 1.** Source data for *Figure 6*.

**Figure supplement 1.** Permutation test results for recordings in the dorsomedial prefrontal cortex (dmPFC) and ventral tegmental area (VTA) for unpunished punishment risk task (PRT) trials after saline or diazepam (2 mg/kg) pretreatment.

supplement 1B,C; *Table 1*). These effects, however, were transient because action latency subsequently decreased. Additionally, the reward retrieval latency increases appeared driven by block 1 as they were not overtly observed in risk blocks (*Figure 5C* – right, *Figure 5—figure supplement 1C*, *Table 1*).

Diazepam influenced neural population activity differently in dmPFC and VTA. In the dmPFC, a reduction during the reward epoch was only observed in the safe block, and activity was not different from saline control levels in risk blocks across any epoch (*Figure 6A–C*). In the VTA, diazepam produced a ramping increase in population activity during the peri-action epoch which began just before action execution (*Figure 6D–E*, *Figure 6—figure supplement 1*) in no risk and 6% risk blocks (where diazepam had normalized behavior, *Figure 5B*). Similar VTA ramps using fiber photometry have been correlated with interoceptive goals (*Guru et al., 2020*). The sustained phasic increase in the VTA by diazepam may, therefore, provide a mechanism to explain its ability to enhance the likelihood of action execution under risk.

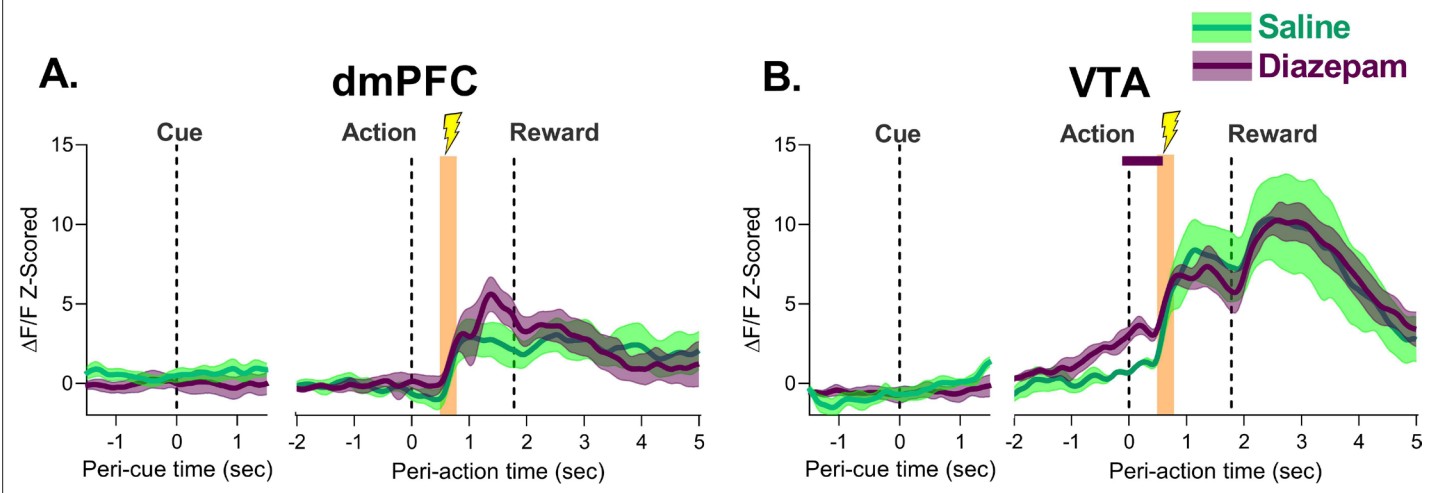

**Figure 7.** Effect of diazepam on neural response to action-contingent footshock in the dorsomedial prefrontal cortex (dmPFC) and ventral tegmental area (VTA). (**A**) dmPFC footshock responses were not different following diazepam treatment. (**B**) VTA response to the footshock did not change with diazepam treatment. The only significant differences were observed during action execution, before the shock was administered. Orange bar indicates period of footshock administration. Traces represent the mean with shaded region indicating ± SEM. Solid lines indicate a significant difference from saline. n=4–7 rats.

The online version of this article includes the following source data and figure supplement(s) for figure 7:

**Source data 1.** Source data for *Figure 7* and *Figure 7—figure supplements 2–3*.

**Figure supplement 1.** Permutation test results for recordings in the dorsomedial prefrontal cortex (dmPFC) and ventral tegmental area (VTA) for punished punishment risk task (PRT) trials after diazepam (2 mg/kg) pretreatment.

**Figure supplement 2.** Area under the curve analysis for each epoch and learning sessions (see *Figure 7—figure supplement 3* for saline/diazepam).

**Figure supplement 3.** Area under the curve analysis for each epoch and saline/diazepam sessions (see *Figure 7—figure supplement 7–S2* for learning data).

Fiber photometry afforded the possibility to assess if diazepam's anxiolytic effects may be related to changes in representation of the footshock. Thus, we separately analyzed the trials which resulted in punishment. Diazepam did not affect the neural response to footshock as both VTA and dmPFC increase neural calcium activity after footshock administration at comparable levels to that of saline (*Figure 7A–B*, *Figure 7—figure supplement 1*). These results suggest that despite being an anxiolytic, diazepam does not change the representation of the punisher by the dmPFC or VTA.

To assess how peri-event signals changed across both learning and diazepam sessions, we quantified AUCs of peri-event changes and compared all three learning sessions and the saline and diazepam treatment sessions together (see *Table 1* for statistics, *Figure 7—figure supplement 2*, *Figure 7—figure supplement 3*). For the dmPFC, no significant change in cue responses was observed over sessions or risk blocks (*Table 1*). For action epochs, dmPFC peri-action phasic decreases were attenuated in 6% and 10% risk blocks over learning and treatment sessions (main effect of risk: see *Table 1*; *Figure 7—figure supplement 2*, *Figure 7—figure supplement 3B*). There was no clear evidence of a global shift between saline or diazepam and learning sessions as no interaction or effect of session was observed. For the reward epoch we observed an increase in calcium activity in the 10% risk block but no effect of session or interaction (*Figure 7—figure supplement 2*, *Figure 7—figure supplement 3C*).

In the VTA we observed a significant increase in the response to the tone cue in the 10% risk block (main effect of risk, *Figure 7—figure supplement 2*, *Figure 7—figure supplement 3E*), but no significant change over sessions or interactions. For VTA action epochs we assessed both difference scores and summation scores because in diazepam sessions both pre and post action activity were increased and thus difference scores masked this effect. For difference scores we observed an effect of session, risk block, and a significant interaction that indicated greater peri-action activity with 10% risk and in Sessions 2–3 and saline (*Table 1*). Post hoc interaction comparisons revealed this was more specifically related to post action increases in the 10% risk block specifically in Sessions 2 and 3

(*Table 1*; *Figure 7—figure supplement 2F*). When taking pre and post action increases into account through summation scores we also observed a significant interaction in the peri-action calcium activity. Post hoc comparisons revealed that diazepam treatment increased summed peri-action activity in the 0% and 6% risk blocks compared to Session 1 (*Figure 7—figure supplement 3F*), but not in Session 2 or 3 (data omitted from *Figure 7—figure supplement 3*). In the VTA a main effect of block and session was observed for the reward epoch, with an enhanced VTA response during the 10% risk block (*Figure 7—figure supplement 2*, *Figure 7—figure supplement 3G*, *Table 1*). Furthermore, reward-related responses were elevated in Session 3 compared to Session 1, though this increase did not continue into saline or diazepam sessions.

No change in footshock responses was observed over sessions in dmPFC or VTA (no main effect of session or interaction). Footshock responses were significantly greater than the corresponding period in trials without footshock (*Figure 7—figure supplement 2*, *Figure 7—figure supplement 3D and H*).

## Correlated activity during action and reward is transiently altered by diazepam

Because PFC co-activity with subcortical regions is critical for reward-motivated behavior and implicated as a potential mechanism of anxiety (*Fujisawa and Buzsáki, 2011*; *Balderston et al., 2017b*; *Sartori and Singewald, 2019*; *Cornwell et al., 2017*; *Park and Moghaddam, 2017*; *Xu et al., 2019*), we asked if diazepam influences the correlated activity of the dmPFC and VTA on a trial-by-trial basis. To assess this, we performed a cross-correlation analysis for all trials after saline or diazepam treatment for the action and reward epochs, an approach which has been used with fiber photometry to identify networks as well as time differences in correlated signals (*Sych et al., 2019*). In saline sessions, correlated activity between the two regions was seen only in the reward epoch with risk. Diazepam increased the correlated activity of dmPFC and VTA during action epochs in the safe and 6% risk block. This increase in correlated activity, however, was not observed when the risk of footshock increased to 10% (*Figure 8A–D*, *Table 1*). In the reward epoch an increase in correlated activity was observed during all blocks (*Figure 8E–H*, *Table 1*). Across all analyses, peak correlations generally appeared with no time lag, except for the action period at high risk, where a 0.5 s VTA lead was observed. Of note, diazepam did not influence the correlated activity during footshock (*Figure 8—figure supplement 1*; *Table 1*).

## Discussion

Learning and adapting to contingencies associated with rewarding or harmful outcomes is critical for survival. In the real world, these outcomes are not independent because actions executed to obtain a reward are often associated with a risk of an aversive outcome. This risk is learned over experience, and the perceived risk of punishment can engender anxiety-related states that will bias action selection. Using a novel task, our previous work demonstrated that the dmPFC and VTA dynamically represent risk of punishment during reward-motivated actions after rats have learned the punishment contingencies associated with these actions (*Park and Moghaddam, 2017*). The present study expands on these findings by focusing on the learning phase of the task, and the impact of diazepam after learning. We find that after exposure to conflicting reward and punishment contingencies, both VTA and dmPFC modify their response in the peri-action and subsequent reward periods but maintain similar responses to the punisher itself. Treatment with diazepam after learning did not influence the neural response to the punisher in either region but it potentiated the VTA action response and enhanced its correlated activity with the dmPFC. These data suggest that action, but not punishment, encoding is critical to adapting behavior to punishment risk contingencies and the resulting state of anxiety.

### Learning of punishment risk produces changes in action, but not punisher, encoding in the dmPFC and VTA

Learning was evident because the suppressive effects of footshock risk were specifically seen for the risky action, but not reward retrieval, in risk blocks after training (*Bolles et al., 1980*). This work provides a substantial expansion of our prior paper because utilizing fiber photometry allowed us to

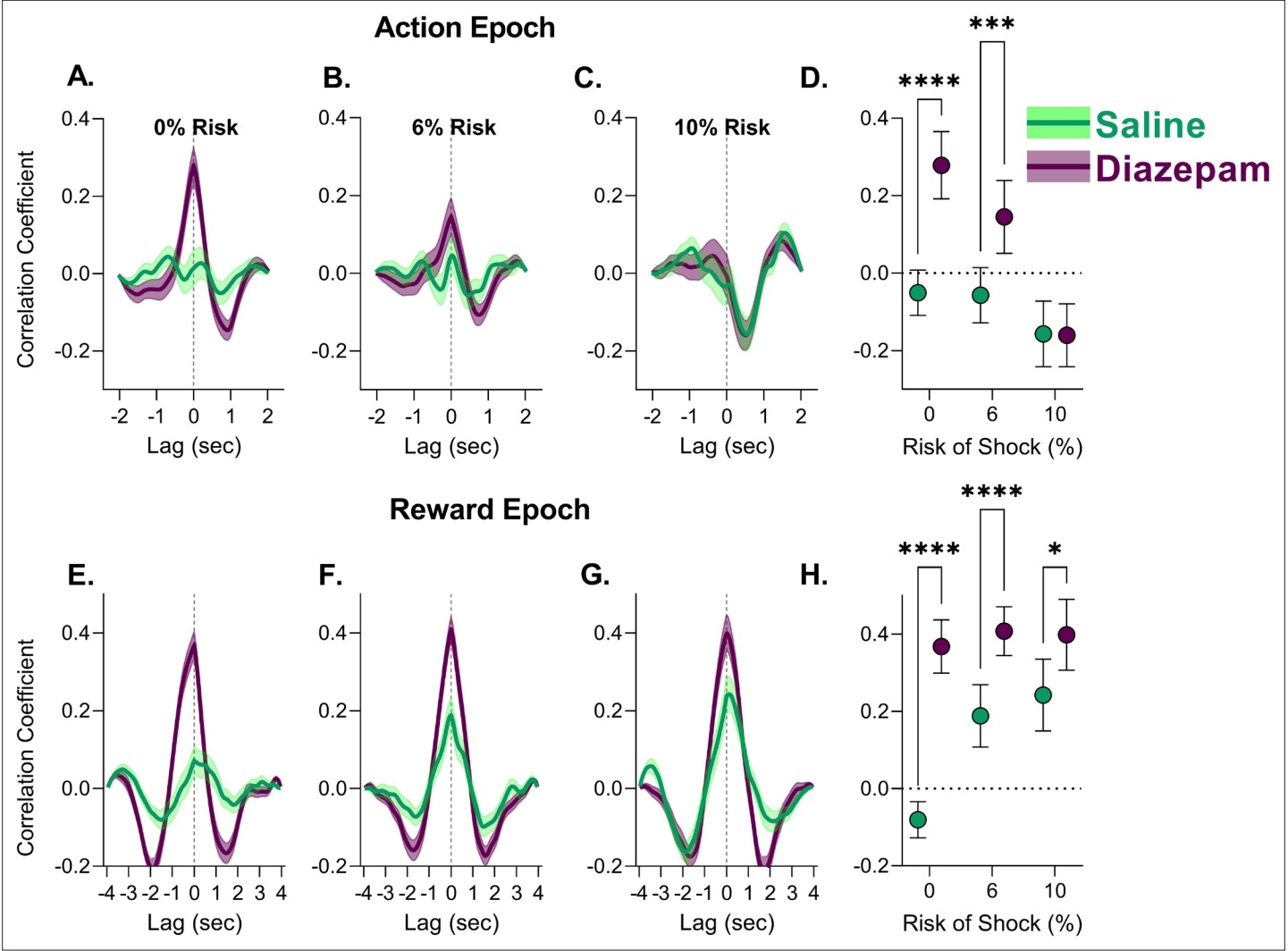

**Figure 8.** Correlated activity between the dorsomedial prefrontal cortex (dmPFC) and ventral tegmental area (VTA) during action and reward epochs in the punishment risk task (PRT) after saline or diazepam treatment. (**A.–C**) Correlated activity during action execution was enhanced by diazepam treatment in the safe block and the lower risk block. While correlated activity reached its lowest level at the highest risk block, regardless of treatment. (**D**) Peak correlation coefficient values and 95% confidence interval for each cross-correlation function in A–C. (**E–G**) Correlated activity was enhanced by diazepam during the reward epoch across all blocks. (**H**) Peak correlation coefficient values and 95% confidence interval for each cross-correlation function in D–F. *p<0.05,****p<0.001. Lines represent the mean with shaded region indicating ± SEM. n=77–120 trials from four rats.

The online version of this article includes the following source data and figure supplement(s) for figure 8:

**Source data 1.** Source data for *Figure 8* and *Figure 8—figure supplement 1*.

**Figure supplement 1.** Cross-correlation results for dorsomedial prefrontal cortex (dmPFC)-ventral tegmental area (VTA) correlated activity for punished (footshock) trials after saline (green) or diazepam (magenta) pretreatment.

measure neural responses during the footshock punisher over PRT training. The change in behavior was not related to dmPFC or VTA responsivity to the punisher itself as the magnitude of footshock encoding did not change in either region during learning. The large phasic response of these regions to footshock is consistent with previous literature implicating these regions in stress and pain responsiveness (*McKlveen et al., 2019*; *Holly and Miczek, 2016*) including our previous work showing that about 50–75% of spontaneously active PFC units respond to stress (*Jackson and Moghaddam, 2006*; *Del Arco et al., 2020*). Previous work has also shown that repeated exposure to the same stressor produces sensitization to some responses such as dopamine release in the prefrontal cortex (e.g. *Gresch et al., 1994*) and rapid desensitization of unit responses in the PFC (*Jackson and Moghaddam, 2006*). We, however, found that repeated exposure to footshock in the context of action

contingent punishment did not produce a sensitized (or desensitized) response. This suggests that context matters in how dmPFC neurons respond to an aversive stimulus. While dmPFC neurons may adapt by reducing their response to a repeated punisher/stressor when stress presentation is certain or not contingent on an action, they appear to sustain the same level of activation when punisher presentation is uncertain and associated with an action.

In contrast to punishment responses, peri-action responses changed in dmPFC and VTA as animals learned the PRT. This finding suggests that action encoding by the dmPFC and VTA can serve as a locus for learning of punishment contingencies during reward-guided behavior. This is an important observation because behavioral studies have shown that adaptation to reward and punishment is mostly associated with the learning of safe and/or dangerous contingencies rather than arousal from punishment or reward (*Jean-Richard-Dit-Bressel et al., 2019*). The neural mechanisms which support reward-punishment contingency learning are thought to be diverse (*McNaughton and Corr, 2004*), and only recently have studies provided direct neural data for such processes (*Jean-Richard-Dit-Bressel et al., 2021b*). Our findings in the dmPFC and VTA implicate these regions in reward-punishment contingency learning and identify action encoding as a key behavioral event involved in this form of learning.

It is interesting that learning the FR1 without punishment also led to the emergence of a phasic peri-action decrease. We posit that this phasic decrease may represent PFC entering an offline state when actions become learned or action outcomes become predictable (see *Wu et al., 2004*; *Kupferschmidt et al., 2017*). Attenuations of this inhibitory response in the PRT could similarly reflect the PFC shifting back to an online state when uncertain and risky contingencies emerge.

Neuronal response to reward in dmPFC and VTA also changed in some phases of PRT suggesting that adaptive responses to reward processing may play a role in the learning of reward-punishment contingencies.

## Diazepam increases peri-action responsiveness of the VTA and VTA-dmPFC correlated activity without influencing punisher and reward responding

Diazepam is a common anxiolytic drug that is used to validate anxiety assays and attenuates the action suppression seen from punishment risk in this task and others (*Jacobs and Moghaddam, 2020*; *Liljequist and Engel, 1984*; *Park and Moghaddam, 2017*). The mechanism for diazepam's anxiolytic effects are incompletely understood (*Sartori and Singewald, 2019*). One possibility is that diazepam itself attenuates responses to anxiogenic stimuli such as punishments. This mechanism was not supported by our data in that we did not see attenuation of footshock responsivity in the VTA or dmPFC after diazepam. An alternative explanation is that diazepam enhances responsivity to reward, which would consequently drive reinforced behavior under punishment risk. Again, our results did not support this explanation, as neural response to reward was similar across regions after drug treatment. Thus, diazepam has little impact on the processing of aversive or appetitive emotional stimuli in these regions. In contrast, diazepam influenced VTA activity during the peri-action epoch by producing a ramping of neural calcium activity where activity rose gradually over the 1.5 s before action in the first two blocks. The so-called 'ramping activity' of VTA neurons has been previously associated with attentional tuning, movement kinetics, and distance to goals (*Kremer et al., 2020*; *Totah et al., 2013*). A recent study which elegantly characterized VTA ramps using fiber photometry found that these signals reflect interoceptive goals particularly when internal maps, and not external stimuli, are utilized to process reward proximity (*Guru et al., 2020*). One possible explanation for our results is that anxiogenic contingencies render subjects more attentive to stimuli and external conditions. Thus, diazepam's production of VTA ramping activity may be a mechanism to direct attentional processes to serve internalized goal-driven states, and ultimately, more efficient reward seeking behavior. This interpretation is also in line with studies which assess the cognitive effects of diazepam in humans, as diazepam has been shown to attenuate vigilant-avoidant patterns of emotional attention to fearful stimuli (*Pringle et al., 2016*).

While diazepam had negligible effects in the dmPFC, it influenced correlated activity between this region and VTA during action epochs in the safe and 6% risk block. This pattern of increased correlation of activity corresponded to the ramping signal in the VTA suggesting that a potential downstream effect of VTA's response to diazepam is to increase its correlative activity with the dmPFC. Correlated

activity between PFC units or local field potentials (LFPs) with subcortical regions including the VTA has been observed during other goal-directed behaviors including PRT (*Fujisawa and Buzsáki, 2011*; *Xu et al., 2019*; *Park and Moghaddam, 2017*; *Mininni et al., 2018*). Thus, while diazepam may not change the activity of dmPFC neurons, it may produce some of its effects by altering correlative activity between dmPFC and VTA. Future investigations using unit and LFP recordings will be needed to better characterize the mechanism for this change in correlated activity under anxiolytic treatment.

Effects of diazepam on behavior and VTA lessened by the last and the highest risk block, when anxiety is presumably highest. Diazepam's half-life is about 1 hr (*Friedman et al., 1986*) and it is possible that this observation is due to lower receptor occupancy by the third (highest risk) block. Another possibility is that diazepam may function in different capacities when threats are either distal or lower in likelihood, that is, blocks 1 and 2, compared to when the threat probability is higher or certain. Future studies could address these possibilities through longer acting anxiolytics or temporally specific manipulations to systematically disrupt the transient and longer lasting effects from diazepam observed here.

## Relationship of fiber photometry data to previous electrophysiology findings

Understanding similarities across neural measurements is an important area of future work as we resolve measurements across different assays and species (*Kriegeskorte et al., 2008*). Fiber photometry signals are not a direct measure of spiking seen in single unit recording, and additional studies investigating its exact relation to single unit approaches are nascent (see *Sych et al., 2019*; *Legaria et al., 2021*; *Xu et al., 2021*). While responses to the cue did not correlate well between the present work and *Park and Moghaddam, 2017*, our observations that dmPFC action-related activity was most sensitive to risk of punishment and the VTA developed phasic increases at the time of action execution are consistent with unit recordings measured in *Park and Moghaddam, 2017*. Similarly, we observed a large reward response in the VTA consistent with previous electrophysiology findings (*Park and Moghaddam, 2017*; *Watabe-Uchida et al., 2017*). Taken together these results provide support that calcium activity of dmPFC and VTA neurons through fiber photometry shows some similarities to the overall population responses seen from in vivo unit recording in this task.

## Conclusion

Assessing how the brain encodes reward-directed actions when conflicted by punishment probability is a novel approach to model learned anxiety. Using an animal model to study this form of learning, we find that dmPFC and VTA selectively adapt action and reward response during the learning of anxiogenic contingencies without modifying their response to the punisher. We also identify peri-action ramping of VTA activity as a potential marker for anxiolytic properties of diazepam in this model. The VTA ramping signal has been linked to preparatory attention and goal-driven states. Potentiation of this response may lead to attentional disengagement from harm probability and provide a mechanism for how diazepam influences action execution under anxiety.

## Acknowledgements

The authors thank Michelle Kielhold for assistance with immunostaining and Alina Bogachuk for help with microscope imaging. DSJ is a recipient of ARCS foundation scholar award.This work was supported by PHS awards from the National Institute of Mental Health R01-MH115026 (BM) and the National Institute of Drug Abuse T32-DA007262 (DSJ, MCA).

## Additional information

### Funding

| Funder | Grant reference number | Author |
| --- | --- | --- |
| National Institute of Mental Health | MH115026 | Bita Moghaddam |

| Funder | Grant reference number | Author |
|---|---|---|
| National Institute on Drug Abuse | DA007262 | David S Jacobs<br>Madeleine C Allen |

The funders had no role in study design, data collection and interpretation, or the decision to submit the work for publication.

### Author contributions

David S Jacobs, Data curation, Formal analysis, Methodology, Writing - original draft, Writing – review and editing; Madeleine C Allen, Data curation, Methodology, Writing – review and editing; Junchol Park, Conceptualization, Formal analysis, Writing – review and editing; Bita Moghaddam, Conceptualization, Resources, Supervision, Funding acquisition, Writing - original draft, Project administration, Writing – review and editing

### Author ORCIDs

David S Jacobs http://orcid.org/0000-0003-3560-7307
Junchol Park http://orcid.org/0000-0002-4739-0793
Bita Moghaddam http://orcid.org/0000-0002-5205-417X

### Ethics

All experimental procedures were approved by the OHSU Institutional Animal Use and Care Committee (Protocol #: 00000727) and were conducted in accordance with National Institutes of Health Guide for the Care and Use of Laboratory Animals.

### Decision letter and Author response

Decision letter https://doi.org/10.7554/eLife.78912.sa1
Author response https://doi.org/10.7554/eLife.78912.sa2

## Additional files

### Supplementary files

• MDAR checklist

### Data availability

Data generated for analyses has been deposited on Dryad. Source code for analysis is available on github (https://github.com/MoghaddamLab/Jacobs2022-eLife, copy archived at swh:1:rev:59d04757f78b43e9b668c1f7276f102b1582d759).

The following dataset was generated:

| Author(s) | Year | Dataset title | Dataset URL | Database and Identifier |
|---|---|---|---|---|
| Jacobs D, Allen M, Park J, Moghaddam B | 2022 | Jacobs-eLife2022-Research Advance | https://dx.doi.org/10.5061/dryad.9s4mw6mkn | Dryad Digital Repository, 10.5061/dryad.9s4mw6mkn |

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

# Appendix 1

## Appendix 1—key resources table

| Reagent type (species) or resource | Designation | Source or reference | Identifiers | Additional information |
|---|---|---|---|---|
| Recombinant DNA reagent | AAV8 GCaMP6s-P2a-tdTomato | OHSU Vector Core | | |
| Chemical compound, drug | Diazepam | Hospira | Product number: 00409-1273-32 | |
| Other | Photometry Recording system | Neurophotometrics | Model FP3001 | Recording system for fiber photometry |
| Other | Optical fibers and patchcords | Doric | Diameter: 400 µm; NA: 0.37; fiber length: 4–8.5 mm | Fibers used for implantation in brain and patchcords for light delivery/detection |
| Other | Arduino | Arduino | Arduino Uno | Microcomputer to interface TTL signals with bonsai |
| Antibody | Anti-GFP (rabbit polyclonal) | Abcam | Catalogue# 6556 | Dilution: 1:500 |
| Antibody | Alexa488 (goat anti rabbit) | Abcam | Catalogue# 150081 | Dilution 1:2000 |
| Software, algorithm | Coulbourn Instruments | Harvard Apparatus | | |
| Software, algorithm | Bonsai | *Lopes et al., 2015* | | |
| Software, algorithm | GraphPad Prism | GraphPad | | |
| Software, algorithm | Zeiss Zen Blue | Carl Zeiss | | |
| Software, algorithm | R, Rstudio | R Project for statistical computing | | |
| Software, algorithm | Python | Anaconda | Version 3.7 | |

