## [Editor Report]

Punishment is key form of learning and behavior change, yet its core behavioral and brain mechanisms remain poorly understood and certainly less understood relative to reward learning. This manuscript uses dual fiber photometry to make an important advance in understanding how punishment is learned by studying how punishment changes action and punisher coding in the medial prefrontal cortex and ventral tegmental area of rats. The authors interpret the results as supporting a role for both areas in foraging in the face of risky outcomes. This work follows nicely on prior work and presents a straightforward and interesting experiment, using a validated anxiolytic to test what components of the neural response are related to this emotional component.

---

## [Decision Letter]

**Decision letter after peer review:**

Thank you for submitting your article "Learning of probabilistic punishment as a model of anxiety produces changes in action but not punishment encoding in the dmPFC and VTA" for consideration by *eLife*. Your article has been reviewed by 3 peer reviewers, including Geoffrey Schoenbaum as Reviewing Editor and Reviewer #1, and the evaluation has been overseen by Kate Wassum as the Senior Editor. The following individual involved in the review of your submission has agreed to reveal their identity: Gavan P McNally (Reviewer #3).

Essential revisions:

While there was general consensus that the paper built nicely on the prior work and represented a meaningful and important Research Advance on that work, there was some disagreement in the reviews over whether the data was appropriately/clearly analyzed and what to do about it. Rather than trying to resolve this into specific points, the authors should look at the reviews and see various suggestions that highlight the respective difficulties the reviewers each had. Generally, there needs to be a bit more clarity and perhaps detail in the analysis of the behavioral and photometric data. At least two reviewers had difficulty locating data supporting or following the logic of data-specific claims made in the text. Additionally, some concern was raised that more discussion or reference should be made to places where the bulk calcium signal diverges from or fails to show information known from the prior report to be in the spiking activity. This came up in the discussion and seems particularly important in the context of this format.

We ask that you please address each of the reviewers points from both the public review and recommendations for authors. If you have not already done so, please include a key resource table and ensure your manuscript complies with the *eLife* policies for statistical reporting: https://reviewer.elifesciences.org/author-guide/full "Report exact p-values wherever possible alongside the summary statistics and 95% confidence intervals. These should be reported for all key questions and not only when the p-value is less than 0.05."

*Reviewer #1 (Recommendations for the authors):*

Suggestions:

1) To address the design issue and the language, I think the authors should just steer clear of strong language about punishment contingencies. As noted above, it is unclear to me whether the behavioral changes reflect the real-time imposition of the shock, or if the rats have really learned specific contingencies. There is some change across sessions, but it seems to me this could be just sensitization to the shock. Or some form of specific learning as the authors suggest. But with so little experience and such sparse shock, it is hard to know exactly. It would be simpler just to refer to what is operationally happening.

2) With regard to the neural data, I think an analysis that compares epochs both within and across the blocks and sessions. This might require approaching the signal differently so that it can be analyzed using multi-factorial approaches. The line plots are great, but supplementing them with bars or scatters that show the values being compared statistically would be great. And I think the diazepam data should be referenced back to these effects. If the shock introduction causes a shift one way, it would be interesting to know if the diazepam transiently reverses this, not just how it changes activity compared to the control in Figure 5.

*Reviewer #2 (Recommendations for the authors):*

– The behavioral results require more rigor and clarity. Whatever the authors choose to perform (mixed effects, ANOVA and/or MANOVA) more thorough descriptions of analysis factors and results for main effects and interactions must be provided.

– In places, the methods lack sufficient detail to understand what was done. I could not determine the full design of the diazepam study. Further, I could not tell which study used Long Evans vs. Sprague-Dawleys, nor why different strains were used for the two studies.

– Both studies are underpowered and did not include sex as a factor in analyses. "Activity" patterns that should have been equivalent on shock and no-shock trials were not so. This makes any apparent differences difficult to interpret. Significantly increasing the sample size is required to detect meaningful differences between calcium transients between sessions and groups.

– It would have been much more of an advance to image only GABA or only dopamine neurons in the VTA. This also would have maximized the strength of the calcium imaging approach. The argument that fluorescence and firing are tightly linked is not convincing.

*Reviewer #3 (Recommendations for the authors):*

– Probably the biggest overall issue is that it is unclear what is being learned specifically. There is no probe test at the end to dissociate the direct impact of shock from its learned impact. And the blocks are not signaled in some other way. And though there seems to be some evidence that the shock effects get more pronounced with a session, it is not clear if the rats are really learning to associate specific shock risks with the particular trials. Indeed with so few sessions and so few actual shocks, this seems really unlikely, especially since without an independent cue, the shock and its frequency is the cue for the block switch. It seems especially unlikely that there is a strong dichotomy in the rats model of the environment between 6% and 10% blocks. This may be quite relevant for understanding foraging under risk. But I think it means some of the language in the paper about contingencies and the like should be avoided.

– The second issue I had was that I had some trouble lining up the claims in the results with what appeared to be meaningful differences in the figures. Just looking at it, it seems to me that VTA shows higher activities at higher shocks, particularly at the time of reward but also when comparing safe vs risky anyway for the cue and action periods. DmPFC shows a similar pattern in the reward period. […] But these results are not described at all like this. The focus is on the action period only and on ramping? I don't really see ramping. it says "Anxiogenic contingencies also did not influence the phasic response to reward...". But fig 3 seems to show clearly different reward responses? The characterization of the change is particularly important since to me it looks like the diazepam essentially normalizes these features of the response. This makes sense to me […].

– I did struggle to understand the functional significance of the PFC transients. I am convinced they are real and robust because we see precisely the same in our own unpublished work. But, I am still puzzled as to what a loss of an 'inhibitory' transient around the punished action in PFC means? This is not really addressed but it is the main effect of punishment on action coding in the PFC and I think some readers would appreciate the author's interpretation of this.

– Re: analyses. I thought these were generally well done. There are two questions one might be interested in. The first is whether the transients are different from 0%. The second is whether transients differ across sessions. The figures do a good job at answering the second question (which to me is the most important question) by using coloured bars above transients to show when session differences are present as assessed by a robust analysis. However, I do think some readers would also appreciate knowing whether and when transients themselves were significantly < or > 0%. Perhaps these figures could be presented as supplementary data.

---

## [Author Response]

Reviewer #1 (Recommendations for the authors):Suggestions:1) To address the design issue and the language, I think the authors should just steer clear of strong language about punishment contingencies. As noted above, it is unclear to me whether the behavioral changes reflect the real-time imposition of the shock, or if the rats have really learned specific contingencies. There is some change across sessions, but it seems to me this could be just sensitization to the shock. Or some form of specific learning as the authors suggest. But with so little experience and such sparse shock, it is hard to know exactly. It would be simpler just to refer to what is operationally happening.

Neural data does not support sensitization to shock. Regardless, we have toned down the use of the term “contingency learning” throughout the manuscript.

2) With regard to the neural data, I think an analysis that compares epochs both within and across the blocks and sessions. This might require approaching the signal differently so that it can be analyzed using multi-factorial approaches. The line plots are great, but supplementing them with bars or scatters that show the values being compared statistically would be great. And I think the diazepam data should be referenced back to these effects. If the shock introduction causes a shift one way, it would be interesting to know if the diazepam transiently reverses this, not just how it changes activity compared to the control in Figure 5.

We have addressed this recommendation in two ways. First we have included permutation analysis comparing to the first block (0%) for each of the first three sessions (also relevant to comments from Reviewer 3; see Figure 2 – supplements 2-3).

Second, to allow comparison within and across sessions, we calculated AUC values for each subject and assessed these with ANOVA for all blocks and sessions (i.e. learning sessions and saline and diazepam). This allows for visualization of individual variability as well as assessing both risk block and session as factors together. Given that not all subjects received injections of saline and diazepam, we used an ordinary two-way ANOVA (non-repeated measures). These results are shown in Figure 7 supplements 2 and 3, and a description has been added to the results. Please note while all sessions were included in the ANOVA, we split learning and saline/diazepam for plotting purposes because having all sessions and individual data on each panel was overly cluttered.

After these analyses, visual inspection of the neural data suggested changes in response as early as session 1. We, therefore, analyzed neural calcium responses in a new dataset involving FR1 training (in shock-naive animals) and split up the trials by early (1-30), middle (31-60), and late trials (61-90) to see if there were any peri-event changes that could be related to time on task rather than risk. These data further demonstrated that changes in neural response were not related to time on task because peri-action and reward responses were the same over FR1 trials in early, middle and late epochs (see Figure 3 – supplement 1).

Reviewer #2 (Recommendations for the authors):– The behavioral results require more rigor and clarity. Whatever the authors choose to perform (mixed effects, ANOVA and/or MANOVA) more thorough descriptions of analysis factors and results for main effects and interactions must be provided.– In places, the methods lack sufficient detail to understand what was done. I could not determine the full design of the diazepam study. Further, I could not tell which study used Long Evans vs. Sprague-Dawleys, nor why different strains were used for the two studies.

Additional clarification has been added to the Methods regarding statistical approach for behavior. For the sake of transparency, we had included plots showing sessions split by each block whereas statistics related to the right side bar plots where data are collapsed across risk (which was done to minimize effects from ‘missing’ data). We appreciate that this may have caused confusion. In the revised manuscript we specify the exact figure for each statistical result and have added a better description in the methods and updated the statistics (Table 1) with the ANOVA and post-hoc results.

– Both studies are underpowered and did not include sex as a factor in analyses. "Activity" patterns that should have been equivalent on shock and no-shock trials were not so. This makes any apparent differences difficult to interpret. Significantly increasing the sample size is required to detect meaningful differences between calcium transients between sessions and groups.

This was not powered to be a sex-difference study and we included both sexes as mandated by NIH, which funded this work.

– It would have been much more of an advance to image only GABA or only dopamine neurons in the VTA. This also would have maximized the strength of the calcium imaging approach. The argument that fluorescence and firing are tightly linked is not convincing.

The parent paper (Park & Moghaddam, 2017) used unit recording in this task (including reporting data from dopamine and non-dopamine VTA units). We assure the reviewer that we do not claim that fiber photometry is a perfect surrogate for direct recording of neural activity. However, a key question we wanted to answer in this study was whether the response of PFC and VTA to the footshock changes during task acquisition (please see last paragraph of introduction), hence the choice to use fiber photometry. We note in the results and discussion that this approach is not optimal for detecting cue or other rapid responses (see page 15 and 23).

Reviewer #3 (Recommendations for the authors):Probably the biggest overall issue is that it is unclear what is being learned specifically. There is no probe test at the end to dissociate the direct impact of shock from its learned impact. And the blocks are not signaled in some other way. And though there seems to be some evidence that the shock effects get more pronounced with a session, it is not clear if the rats are really learning to associate specific shock risks with the particular trials. Indeed with so few sessions and so few actual shocks, this seems really unlikely, especially since without an independent cue, the shock and its frequency is the cue for the block switch. It seems especially unlikely that there is a strong dichotomy in the rats model of the environment between 6% and 10% blocks. This may be quite relevant for understanding foraging under risk. But I think it means some of the language in the paper about contingencies and the like should be avoided.

While the parent paper (Park & Moghaddam, 2017) delved more deeply into this question we agree that what exactly is learned may be difficult to ascertain. To address this (please also see response to reviewer #1’s first comment), we have toned down our use of the “contingency learning” throughout the manuscript and use the word contingency in relation to the underlying reinforcement/punishment schedules.

The second issue I had was that I had some trouble lining up the claims in the results with what appeared to be meaningful differences in the figures. Just looking at it, it seems to me that VTA shows higher activities at higher shocks, particularly at the time of reward but also when comparing safe vs risky anyway for the cue and action periods. DmPFC shows a similar pattern in the reward period. […] But these results are not described at all like this. The focus is on the action period only and on ramping? I don't really see ramping. it says "Anxiogenic contingencies also did not influence the phasic response to reward...". But fig 3 seems to show clearly different reward responses? The characterization of the change is particularly important since to me it looks like the diazepam essentially normalizes these features of the response. This makes sense to me […].

We initially believed that much of the differences in reward (with the exception of Session 2 in the PFC) were from carryover of differences in the peri-action period. However upon quantifying these responses again using AUC change scores to adjust for pre-event differences in the signal, we observed small reward related increases (data are in Figure 7 – supplements 2/3) and have updated results and the discussion.

Although some lessening of reward response may be apparent across the diazepam session in the VTA (Figure 7 – supplement 2/3G), we do not have statistical support for this as no significant differences were observed in permutation comparisons to saline and only session 3 deviated from the first session for the reward period in the AUC analyses.

I did struggle to understand the functional significance of the PFC transients. I am convinced they are real and robust because we see precisely the same in our own unpublished work. But, I am still puzzled as to what a loss of an 'inhibitory' transient around the punished action in PFC means? This is not really addressed but it is the main effect of punishment on action coding in the PFC and I think some readers would appreciate the author's interpretation of this.

We have added some discussion about this observation. Additionally, we have added new data (now Figure 3) assessing the action related activity from recording when these same animals learned the FR1 response (before any footshock exposure). These data demonstrate that, as the action-outcome association is learned, an inhibitory transient emerges in the PFC. This suggests that a decrease in the inhibitory phasic response may reflect a less engaged PFC state when actions are predictable and learned. Speculation related to this potential mechanism has been added in the Discussion.

Re: analyses. I thought these were generally well done. There are two questions one might be interested in. The first is whether the transients are different from 0%. The second is whether transients differ across sessions. The figures do a good job at answering the second question (which to me is the most important question) by using coloured bars above transients to show when session differences are present as assessed by a robust analysis. However, I do think some readers would also appreciate knowing whether and when transients themselves were significantly < or > 0%. Perhaps these figures could be presented as supplementary data.

To address this comment (and a similar one from reviewer 1) we have included analysis comparing each session to the 0% risk block over the learning sessions. We have also computed AUC analysis for all sessions to show how these changes change compared to 0% shock risk session.